# Evaluation of a “Picture Assisted Illustration Reinforcement” (PAIR) System for Oral Hygiene in Children with Autism: A Double-Blind Randomized Controlled Trial

**DOI:** 10.3390/children10020369

**Published:** 2023-02-13

**Authors:** Atrey J. Pai Khot, Abhra Roy Choudhury, Anil V. Ankola, Roopali M. Sankeshwari, Vinuta Hampiholi, Mamata Hebbal, Sagar Jalihal, Ram Surath Kumar, Laxmi Kabra, Sree Lalita Kotha

**Affiliations:** 1Department of Public Health Dentistry, KLE VK Institute of Dental Sciences, KLE Academy of Higher Education and Research (KLE University), Belagavi 590010, Karnataka, India; 2Department of Periodontics, KLE VK Institute of Dental Sciences, KLE Academy of Higher Education and Research (KLE University), Belagavi 590010, Karnataka, India; 3Department of Preventive Dental Sciences, College of Dentistry, Princess Nourah bint Abdulrahman University, P.O. Box 84428, Riyadh 11671, Saudi Arabia; 4Department of Basic Dental Sciences, College of Dentistry, Princess Nourah bint Abdulrahman University, P.O. Box 84428, Riyadh 11671, Saudi Arabia

**Keywords:** autism-spectrum disorder, behavior, cognition, communication, dental caries, health education, oral hygiene

## Abstract

This study evaluated the potential of a novel pre-validated “Picture Assisted Illustration Reinforcement” (PAIR) communication system and conventional verbal techniques for Oral Health Education (OHE) in terms of dentition status, gingival health, oral hygiene status, and practices in 7- to 18-year-old children with Autism Spectrum Disorder (ASD). A double-blind randomized controlled trial was undertaken in a school for children with autism from July to September 2022. A total of 60 children were randomly assigned into two groups: a PAIR group (*n* = 30) and a Conventional group (*n* = 30). Cognition and pre-evaluation of all the children were assessed by standardized scaling measures. A pre-validated closed-ended questionnaire was administered to caregivers of both groups. At a 12-week post-intervention, a clinical examination was performed using the World Health Organization (WHO) Oral Health Assessment form 2013, gingival and Oral Hygiene Index Simplified (OHI-S). The gingival scores in the PAIR group (0.35 ± 0.12) exhibited a statistically significant decline in scores as compared to Conventional group (0.83 ± 0.37), *p* = 0.043. Meanwhile, the oral hygiene scores in the PAIR group and Conventional group were 1.22 ± 0.14 and 1.94 ± 0.15, respectively (*p* < 0.05). A significant improvement in oral hygiene practices was observed in the PAIR group. Incorporating the PAIR technique resulted in significant progress in child cognitive ability and adaptive behavior, which reduced gingival scores and improved oral hygiene scores, consequently improving oral hygiene practices among children with ASD.

## 1. Introduction

The term “autism” was first coined by Bleuler (1911) to describe a specific abandoning-behavior problem prevalent in schizophrenia patients [1]. Over the past several years, the criteria used to describe autism have changed notably, but the most accepted definition states that “Autism Spectrum Disorders (ASDs) are a group having severe, pervasive neuro-developmental impairment; characterized by a triad of substantial qualitative impairments in reciprocal social interaction and communication, and restricted, repetitive patterns of behavior, interests, and activities” [2]. Children with autism present significant deficits in social interaction skills, such as being inattentive when called by their name, and problems comprehending social cues [3]. In addition, children between 12 and 14 months have difficulties with verbal and nonverbal communication in a manner that is usually noticeable [4]. Approximately 25% of children with autism are unable to employ the natural speech process as their primary mode of communication [4,5]. The current literature advocates the etiology to be multifaceted, involving genetic mutations, neuropsychopathy, exposure to heavy metals, and fetal exposure to medications (valproic acid or thalidomide) [6]. For children with ASD, learning and practicing healthy hygiene behaviors may not be easy and requires repeated oral hygiene instructions and the involvement of caregivers [7]. Studies reported caries prevalence in 21–77% and gingivitis in 62–97% of children with ASD [8]. Additionally, dental care for children with special needs is often neglected due to the paucity of trained dentists specialized in treating such individuals and the inability to manage the behavior of these children [1,9]. Consequently, they are at risk of developing oral diseases due to inadequate oral hygiene practices, dietary habits, damaging oral habits such as bruxism, concurrent medical problems, variable cognition level, altered saliva level, motor coordination deficits, oversensitivity to sensory stimuli, and oral self-injures [10]. The increased number of cases of ASD diagnosed worldwide show that it is imperative to develop tools to educate these children and develop their communication skills [5,11]. Special educators and therapists have opined that early intensive, continuous, and reinforced educational programs and behavioral therapies help them achieve better self-care, social interaction, and communication [5,12].

The Picture Assisted Illustration Reinforcement (PAIR) Communication System was inspired by the Picture Exchange Communication System (PECS), being a modified version of the same that is reliant on descriptive data. It is a novel behavior-based pictorial system built to develop communication skills and educate children with ASD regarding oral hygiene practices to serve as the interlink between daily practices. PAIR intervention is based on the concepts of applied behavior analysis and uses distinctive teaching, reinforcement, and backward-chaining strategies [12]. Additionally, this system employs pictorial differentiation of oral hygiene aids and structured illustrations, which demonstrate oral hygiene tasks in distinct patterns. The lack of uniform assessment procedures for robust experiments in the PECS technique causes difficulties for analysis, which raises the complexity level for implementation [13]. The PECS approach does not exploit the distinction between correct and incorrect references or pictures to bridge a communication message [13,14]. Although PECS has been incorporated in the UK since 1998, its application in preschools and schools of special children remains limited [14]. PAIR integrates an evidence-based process that does not depend on an additional language system or any prerequisite skill requirements such as imitation or intentional abilities. Children with ASD are often visual learners and will, therefore, respond better to visual support rather than written or spoken words [15]. In terms of biologically oriented outcome measures, prior research has indicated that visual patterning may be a viable tool for monitoring modest responses to intervention and determining distinctive elements of the child’s performance that align with the postulated process of change [13,15]. This novel PAIR technique may effectively improve oral hygiene and significantly impact the systemic health and relative morbidity of these children. 

Although there has been substantial advancement in the study of ASD within the past decade, translational research about the efficacy and effectiveness of interventions is hampered by extreme heterogeneity in models, as well as in outcome measures. Hence, the objective of this study was to assess Autism Spectrum Quotient (ASQ), Intelligence Quotient (IQ), and Social Quotient (SQ) and their cognitive association with dentition status, gingival health, oral health status, and oral hygiene practices using the Picture Assisted Illustration Reinforcement (PAIR) communication system as compared to a conventional verbal technique in 7–18-year-old children with ASD before and after the intervention.

### 1.1. Null Hypothesis

There is no difference in cognition level and dentition status, gingival health, oral hygiene status, and practices when OHE is conveyed using a PAIR technique as compared to a conventional verbal technique among 7–18-year-old children with autism spectrum disorders at 12-week intervals.

### 1.2. Alternative Hypothesis

There is a difference in cognition level and dentition status, gingival health, oral hygiene status, and practices when OHE is conveyed using a PAIR technique as compared to a conventional verbal technique among 7–18-year-old children with autism spectrum disorders at 12-week intervals.

## 2. Materials and Methods

### 2.1. Study Design and Study Setting

The study was conducted on children with ASD in the age range of 7 to 18 years who were attending special schools for children with autism in Belagavi, Karnataka, India. The study was executed during the period from July to September 2022 as a double-arm, double-blinded, randomized, and controlled trial. Baseline and intervention sessions were conducted in classrooms in the presence of caregivers. The trial was registered under the Clinical Trials Registry—India with the CTRI number CTRI/2022/06/043555, following Consolidated Standards of Reporting Trials (CONSORT) guidelines.

### 2.2. Ethical Considerations

Ethical clearance was obtained from the Institutional Research and Ethics Committee (IRB number: EC/NEW/2021/2435) with reference number: 1534, dated: 28 March 2022. This study strictly adhered to the ethical standards of human experimentation and the Helsinki Declaration of 1975, revised in 2000.

### 2.3. Pilot Study

A pilot study was conducted on 15 children with autism to determine feasibility, the time required for oral health examination, and data collection, which was excluded from the main study samples. The WHO Oral Health Assessment (2013) was utilized to record the findings. A self-designed 8-item closed-ended questionnaire was prepared and assessed for reliability using Cronbach’s alpha (0.87) and validity using the content validity ratio (0.84).

### 2.4. Sample Size Estimation

The necessary sample size for the study was estimated using the GPower program (G*Power Version 3.1.9.4 statistical software) to be 27 children in each group, accounting for a total sample size of 54 at a power of 0.85 and an alpha error of 0.05 [16]. Hence, assuming a 10% dropout rate, this study included a sample size of 30 children in each group, for 60 children in total.

### 2.5. Inclusion and Exclusion Criteria

We considered children and adolescents in the age group of 7 to 18 years that had been diagnosed with autism spectrum disorder by a multidisciplinary team according to the International Classification of Diseases, 10th Revision (ICD-10) [17] and Diagnostic and Statistical Manual of Mental Disorders (DSM V) [18] criteria. 

*Inclusion criteria*: Children diagnosed with autism spectrum disorder, permanent or mixed dentition, cooperative children, and caregivers of children who assented and signed informed consent to participate in the study. 

*Exclusion criteria*: Children with visual impairment, other special needs or challenges, and underlying systemic diseases; those with Frankl scale score < 2 and uncooperative behavior; and/or other special health care needs, children who have received OHE and treatment sessions in the last 12 weeks. 

### 2.6. Phases of Study Conduct

#### 2.6.1. Preparatory Phase

Standardized psychological and cognitive tests were conducted one month prior to the onset of this study. Examiners were trained to record the WHO Oral Health Assessment form 2013 [19], Oral Hygiene Index Simplified (OHI-S) [20], and Loe and Silness gingival index [21], which was supervised by subject experts. Intra-examiner and inter-examiner reliability scores were (0.82, 0.84) and (0.81, 0.86), respectively, using kappa statistics, which indicated a substantial level of agreement. The sensitivity and specificity of the ASQ as a screening tool for ASD were evaluated using the Receiver Operating Characteristic (ROC) curve. The internal consistency of the three trials for the Raven’s Coloured Progressive Matrices Technique was high, at 0.906. 

#### 2.6.2. Cognitive and Pre-Evaluation Phase

Interactive sessions were conducted to understand the cooperation and cognition of children with ASD. The Autism Spectrum Quotient (ASQ) [22] was assessed by a questionnaire to measure the traits associated with the autistic spectrum under five domains (10 items per domain). All items were assessed on a four-point scale, with one representing strongly agree and four representing strongly disagree. Furthermore, children were evaluated with Raven’s Coloured Progressive Matrices Technique and Vineland Social Maturity Scale (VSMS). Assessments were conducted in a distraction-free setting in a separate field clinic. 

#### 2.6.3. Intervention Phase

A total of 60 children participated in the study. A simple random-sampling technique was employed by using the lottery method. Children were randomly assigned into two groups, a PAIR group (*n* = 30) and Conventional group (*n* = 30), using a computer-generated table of random numbers. Allocation concealment was administered using the SNOSE (Sequentially Numbered, Opaque, Sealed Envelope) technique. The questionnaire, which was provided in the English language, comprised eight practice-based questions and was distributed to the caregivers. The responses, including information such as sociodemographic details and information on oral hygiene practices, were collected by the examiners. This was followed by a clinical examination conducted by two examiners to record WHO Dentition Status (2013), OHI-S, and Loe and Silness gingival index. After the baseline assessment, the investigator delivered OHE to the PAIR group and Conventional group. The type of intervention given by the investigator was masked from the examiners and was blinded from group assignment. To minimize contamination, interventions were administered in two separate schools for both groups.

##### PAIR Technique

This novel technique is based on illustrations that show the sequence of actions required to maintain oral hygiene (Figure 1). The steps included were as follows: 

*Step 1*: Illustration and identification of the pictures by the children. 

*Step 2*: Children were handed the illustrated object in exchange for choosing the correct picture. 

*Step 3*: Children identify both the correct and incorrect illustrations that were displayed. 

*Step 4*: Children frame a sentence following the sequence of the pictures. 

The first set consisted of small laminated sequential pictures (do’s) with textual instructions to be displayed on a Velcro^®^ board (Velcro IP Holdings LLC, Manchester, NH, USA). The second set of sequential pictures (don’ts) consisted of textual instructions on each of the illustrations. The illustrative sets were 7 × 6 cm in dimension, laminated, and arranged in a sequential order to facilitate effective handling and better understanding by the children. These instructions were divided into sub-sections relating to brushing essentials, dental floss, healthy diet, and routine dental visits. The ‘brushing essentials’ subsection demonstrated a stepwise illustration of the Fones method of toothbrushing along with the other list of requisites. 

This was followed by training individual children to mimic the brushing technique on the tooth model, which every child was encouraged to perform until the technique was perfected. Finally, periodic reinforcement of OHE was provided at the 1st week, 4th week, and 8th week in both groups. A group session was held to teach caregivers the steps of PAIR using a PowerPoint presentation and models at the field clinic for two hours. 

##### Conventional Technique

A trained examiner delivered OHE verbal talk to the children of Group B in the presence of school teachers for a better understanding. The content of health education was similar in both the PAIR group and Conventional group. Finally, the Fones method of toothbrushing was demonstrated to the children. A CONSORT flow diagram is depicted in Figure 2.

#### 2.6.4. Post-Intervention at 12th Week

The final assessment of dentition status, oral hygiene status, and gingival scores of all the children in both groups was conducted in the 12th week, using the same indices as the baseline assessment. These indices were recorded in the time period between 11:00 am and 12:30 pm post-breakfast and pre-lunch, with no in-between snack time. Oral hygiene practices were re-assessed using the same questionnaire after 12 weeks to assess the impact of both OHE techniques. The examiners who were blinded during the grouping of the children carried out the examination, thereby minimizing the bias. The control group also received a similar type of OHE (PAIR technique) after the completion of phases. 

The total score of the questionnaire was computed based on each caregiver’s response. Each correct response was scored as “1” and the incorrect response as “0”. The oral hygiene practice scores were categorized into healthy (>50th percentile) and unhealthy (≤50th percentile) [23].

### 2.7. Statistical Analysis

The data obtained were entered in Microsoft Excel 2020 and subjected to statistical analysis by a blinded statistician, using IBM SPSS^®^ Statistics for Windows, Version 21.0 (IBM Corp., Armonk, NY, USA), released in 2012. The descriptive statistics were presented as mean ± standard deviation for continuous variables and as frequencies with percentages for categorical variables. The normality of the distribution of the continuous variables was determined using the Shapiro–Wilk test. As the data were normally distributed (*p* > 0.05), chi-squared analyses were carried out to evaluate the association between the sociodemographic characteristics and the cognitive status of the children. McNemar analysis was carried out to analyze the differences in practice outcomes between the PAIR group and Conventional group before and after the OHE intervention. Paired and unpaired *t*-tests were applied to compare the mean gingival score, oral hygiene scores, and practice scores between both groups. Statistical significance was set at *p* ≤ 0.05.

## 3. Results

### 3.1. Cognition and Sociodemographic Status of the Children

The mean age of the PAIR group and Conventional group children was 11.6 ± 3.01 and 12.07 ± 2.66 years, respectively. Of the 60 participating children, 38 (63.33%) were males, and 22 (36.67%) were females. Both male and female children had a similar age distribution. The analysis showed an average mean intelligence (IQ) of 19.9 ± 6.68, significantly lower than the corresponding mean social maturity score (SQ) of 57.83 ± 6.59. There was no significant difference (*p* > 0.05) in baseline characteristics of ASQ, VSMS, IQ, and Frankl scale in children of both groups. The VSMS, IQ, and Frankl behavior scale in children of different genders showed statistically significant differences (*p* ≤ 0.05). However, ASQ was found to be statistically insignificant concerning gender-wise distribution (*p* > 0.05). Table 1 presents the distribution of children with ASD in this study according to their intervention groups, gender, IQ, and behavior classification profiles.

### 3.2. Oral Health Conditions among the Children

At baseline health, 51.67% of the children had poor oral hygiene, 25% had severe gingivitis, 31.67% had enamel fractures, and 16.66% had ulcerations, and 6.67% had soft tissue lesions. The total mean DMFT scores of the study population were 5.10 ± 0.350. When mean scores of oral hygiene conditions were compared based on IQ and SQ, a significant difference in DMFT scores was noted, with profound retardation having the highest mean DMFT Score with a *p* value of 0.004 (Table 2).

### 3.3. Caregiver Perception of Oral Hygiene Practices among the Children

The McNemar test revealed that oral hygiene practices in the PAIR group and Conventional group had significant effects on the post-intervention scores. The percentage of good oral hygiene practices among children was significantly higher after the intervention (*p* < 0.001). The mean practice score at the 12-week follow-up indicated a statistically significant difference between the groups; *p* < 0.001. Table 3 summarizes the oral hygiene practices in the children of both groups before and after OHE intervention. The unpaired *t*-test revealed that the baseline practices in both groups were almost equal and statistically insignificant (*p* = 0.846). The paired *t*-test showed that the percentage of healthy practices was significantly high after the intervention (*p* < 0.001). Noticeable improvements in oral hygiene practices were observed, with a statistically significant difference between the groups (*p* < 0.001) (Table 4).

The unpaired *t*-test indicated that the baseline gingival and oral hygiene scores in both groups were almost equivalent and statistically insignificant (*p* > 0.05). Paired *t*-test revealed both indices indicating a statistically significant reduction in the gingival and oral hygiene scores in either group from the baseline to 12 weeks interval (*p* < 0.001). The gingival scores in the PAIR group (0.35 ± 0.12) exhibited a statistically significant decline in scores as compared to the Conventional group (0.83 ± 0.37) with *p* = 0.043 (Figure 3). Meanwhile, the oral hygiene scores in the PAIR group and Conventional group were 1.22 ± 0.14 and 1.94 ± 0.15, respectively, with statistically significant differences between the groups found by unpaired *t*-test (*p* < 0.05) (Figure 4). The baseline and post-intervention comparison practice scores in the PAIR group and Conventional group are represented by a violin plot (Figure 5).

## 4. Discussion

Autism Spectrum Disorders (ASD) have been receiving increased attention over the last decade as awareness and knowledge about them grow. Our study was motivated by the higher risk of oral health disparities that special children face as compared to their counterparts [24]. This has resulted in poor oral health and is mainly due to difficulty in sensory processing, difficulty in communication, erratic behavior, and inaccessibility to professional services [25]. The oral health status of children with ASDs is a matter of concern, and appropriate measures must be taken to prevent it from deteriorating. ASD children have problems locating dental practitioners willing to provide dental treatment, limiting accessibility to treatment modalities [26]. Dentists around the world agree that they did not receive adequate training during their curriculum to work with this particular population. One way of dealing with the predicament is to introduce the “Patient-Centered Model of Care”, which exposes dentists to a wide diversity of patients, including ASD children, and also helps them acquire hands-on experience [27]. Furthermore, studies have reported that tooth-brushing is a cumbersome process for ASD children, which can be exacerbated by the sensation of bristles in the mouth or even the texture of the toothpaste. Similarly, 61% of parents have difficulty training their children to maintain their oral hygiene [28]. The primary purpose of this study was to educate such children, with the help of the PAIR technique, to be self-reliant in maintaining their oral hygiene. 

Dental treatment can be unpleasant for both children with ASD and healthcare workers, which further emphasizes the significance of catering to their needs [29,30]. Learning about important measures for establishing high standards of dental education and training to assess and treat children with special needs is a step in the right direction to tackling these issues [31]. Despite several enthusiastic efforts, improvement in the oral health of ASD children is agonizingly slow, especially in countries like India. According to a study conducted by Pini et al. [32], these children’s Decayed–Missing–Filled teeth (DMFT) index was relatively higher than neurotypical children. Similar results were observed in the current study, with the majority of the children having a poor oral hygiene status with a DMFT score of 5.10 ± 0.350. The higher incidences of such problems can be attributed to the fact that adequate oral hygiene maintenance is difficult for these children, and other factors like the presence of mouth-breathing, adverse effects of medicines, and occlusal abnormalities might compound the issues even further. This further clarifies the need to focus on oral health education and behavior-shaping methods as preventive and non-invasive measures towards improving oral hygiene status [32]. According to a study by Mehta et al. [33], immediate care programs and planning are required due to the poor oral health of children with special care needs. This led us to the formulation of our new PAIR technique, which has the potential to sculpt a niche for itself among extant effective oral health education methods for children with autism through a series of dynamic, interactive, visually appealing illustrations. 

The Intelligence Quotient (IQ), as well as the Social Maturity score (SQ) of the ASD children involved, were relatively low; this result was consistent with a similar evaluation conducted in another research setting [34]. Low IQ and SQ can limit adaptive behaviors significantly, which induces anxiety towards performing new or unfamiliar tasks. Moreover, the associated psychiatric concerns have an early onset, persist for a lifetime, and cause a higher level of impairment that increases rapidly in these children, making them a major concern for parents and caregivers. A meta-analysis was conducted by Edirisooriya et al. [35] to establish a correlation between IQ and the development of internalizing symptoms such as anxiety among ASD children, which established a negative relationship. Children with lower IQs have poor coping abilities, especially while executing some unfamiliar task. IQ also has a bearing on the ability to communicate and express one’s feelings, which results in these children becoming more unresponsive. Studies have also focused on alleviating anxiety in special children to positively impact their IQ and SQ. The best example of this approach is the use of an Autism Hug Machine Portable Seat (AHMPS) in a study by Afif et al. [36]. In this study, a deep-pressure integration therapy that applied adequate pressure at stimulating areas of the body to induce the autonomic nervous system to induce a sense of calm was used, and the results were quite promising. The heart rate was reduced and the conductance of the skin improved, making it an effective tool for relaxation and dispersion of anxiety. Another study by G. Lefer et al. [37] emphasized the use of visual pedagogy techniques to acclimatize children to dental procedures so that they become more receptive to them. A digital application called çATED secured the cooperation of the children to a large extent. As the age of the children increased, the number of them with adequate intellectual ability also increased. In terms of gender, a highly significant association was noticed in the current study with IQ and VSMS. Although a direct causal relationship cannot be developed between gender, intelligence, and social maturity, males are less protected than females for the manifestations of autism, and genetic liabilities may have a role to play in it. Lower IQ and SQ were associated with low verbal expression abilities and difficulties with the expression of fear, which led to the internalizing of these symptoms. This was tied to the formation of deep-seated emotional issues that grow and are carried into adolescence and later stages of life. 

A detailed look at the previous literature revealed that children with autism had poor oral hygiene and an increased incidence of dental pathology even when they had normal development [38]. A study by Morales-Chávez et al. [39] reported that young children with autism between the ages of 2–16 years had high caries prevalence as compared to other normal children in the same age group. After evaluating the gingival status of the children, it was observed that 83.3% presented with gingivitis. A similar observation was reflected in the current study: the majority of children presented with gingivitis and poor oral hygiene. One of the basic and most important strategies for improving the oral hygiene of children with ASD is teaching them appropriate brushing techniques. The PAIR technique strives to accomplish this, and in the present study, we observed a marked reduction in terms of gingival and oral hygiene scores. The post-intervention analysis revealed higher oral hygiene practice scores in PAIR group as compared to the Conventional group with lower oral hygiene practice scores. Thus, the PAIR technique is an effective oral health education technique for children with ASD. In the present study, children presented with enamel fractures due to trauma, which can be associated with comorbidity of autism in children in the absence of muscular coordination, as supported by Marra et al. [40]. The altered tone of the muscles and the higher incidence of flaring of the incisors predisposes the children to injury even further. 

The PAIR technique aims to improve comprehension and social communication skills among children with ASD. Intelligent verbalization is implemented through an evidence-driven procedure in this technique and does not require the learning of an additional language or skill. The PAIR technique is based on two aspects, effective communication and using visual cues to stimulate learning, with a special emphasis on reinforcement. This combination of behavioral and functional sequence analysis substantially enhances its utility. Providing regular and informative training is important, and the current study promotes educating caregivers about the execution of the PAIR technique. Moreover, ASD children find it extremely difficult to communicate with dentists and remember the instructions that are given, necessitating continuous reinforcement of the technique [41]. The PAIR technique takes all of the aforementioned aspects into consideration and thus has the potential to become an effective method for providing oral health education to ASD children.

### 4.1. Strength and Limitations

The study was balanced by selecting children with standardized baseline scores in both groups. Many similarities were present between the two groups of children receiving oral health education in terms of their demographic variables as well as the pre-treatment measures of the outcome. This helped to maintain a post-treatment specificity to the interventions involved. Children were characterized based on cognitive level, age, and behavior pattern to understand the barriers in communication.

This novel approach of conveying OHE using the PAIR technique offers an effective intervention option for health care professionals that enhances rapport-building and piques the interest of children with ASD, thereby aiding in the development of positive oral hygiene practices. Our approach has a further advantage in being an OHE tool that can be employed repetitively at no additional expense or training of health professionals.

The limitation of the study is the short follow-up period. It was obligatory to augment assistance from caregivers throughout the study. Children with autism are known to have shorter memories and attention spans. Therefore, regular follow-up and periodic reinforcement will provide a more comprehensive view of the viability of the PAIR technique. 

### 4.2. Clinical Significance 

Specific OHE techniques catering to populations with special needs, in the long run, have a long-lasting impact on our society. Hence, the PAIR technique can be used as a fundamental health educational tool to increase awareness among children with autism. It is a simple, effective alternative technique that employs picture illustrations and engaging, interactive sessions to provide oral hygiene instructions that can be implemented by all healthcare professionals in day-to-day practice. 

## 5. Conclusions

Incorporating the PAIR technique revealed that early intensive intervention leads to significant progress in children’s cognitive ability and adaptive behavior, making it an effective tool for reducing gingival scores, improving oral hygiene scores, and consequently improving oral hygiene practices among children with ASD. The incorporation of the PAIR technique in OHE programs is easy and serves as an appealing, interactive, and cost-effective method. Children with autism commonly present poor oral hygiene and gingival conditions, which increase the risk of developing dental diseases. Their behavior and life factors may complicate the provision of services and limit access to dental care. Thus, implementation in clinical settings will assist clinicians in making informed decisions for children with ASD. 

Longitudinal studies involving several health education sessions between children, teachers, parents, and healthcare professionals can be conducted to corroborate the outcomes of this study. These findings can be extrapolated to children of all ages, socioeconomic backgrounds, and schools in different geographic regions. The results from such research would inform the extent of its generalization to other settings, such as preschool children, by optimizing the intervention with input from expert consultants. In the future, this technique might be an essential tool and medium of instruction for autistic special schools, resulting in a substantial improvement in the oral health of future generations.

## Figures and Tables

**Figure 1 children-10-00369-f001:**
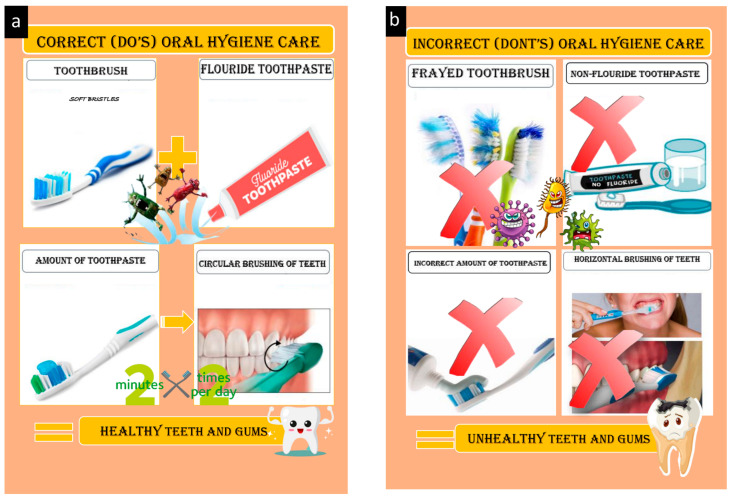
Picture illustrations; (**a**) correct (do’s) oral hygiene care; (**b**) incorrect (don’ts) oral hygiene care.

**Figure 2 children-10-00369-f002:**
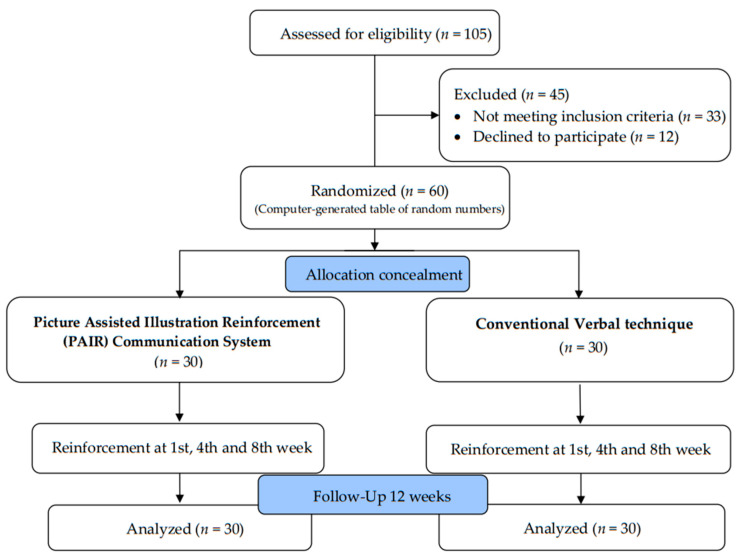
Consolidated Standards of Reporting Trials (CONSORT) diagram.

**Figure 3 children-10-00369-f003:**
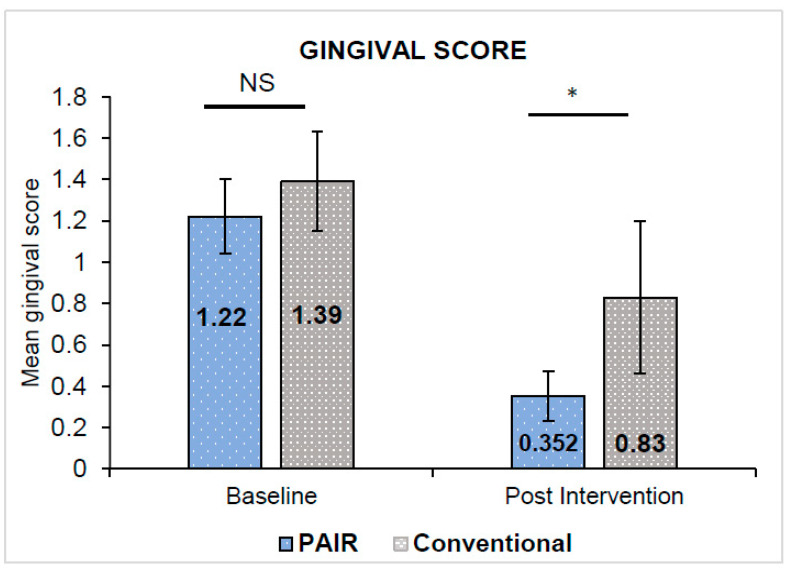
Comparison of mean gingival score in PAIR technique group and conventional technique group. All values are expressed as mean ± standard deviation. The statistical test used: Unpaired *t*-test; * Statistically significant, *p* ≤ 0.05; NS: not significant.

**Figure 4 children-10-00369-f004:**
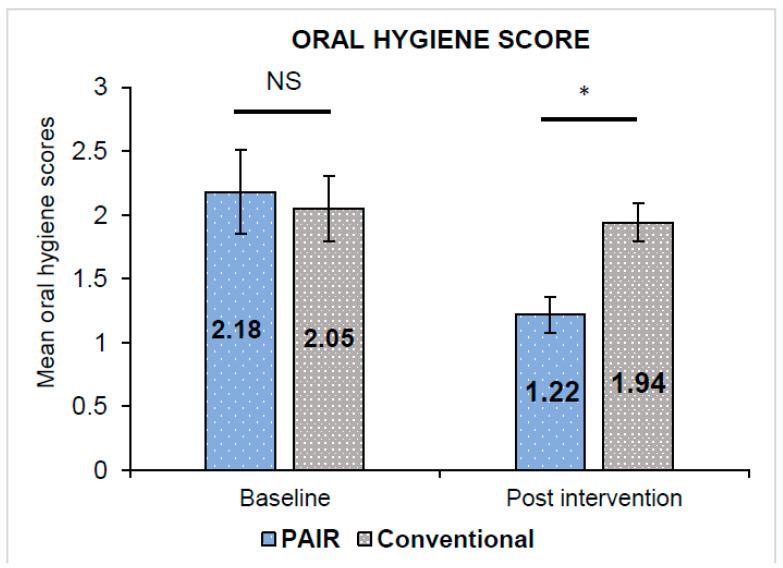
Comparison of mean oral hygiene score in PAIR technique group and conventional technique group. All values are expressed as mean ± standard deviation. The statistical test used: Unpaired *t*-test; * Statistically significant, *p* ≤ 0.05; NS: not significant.

**Figure 5 children-10-00369-f005:**
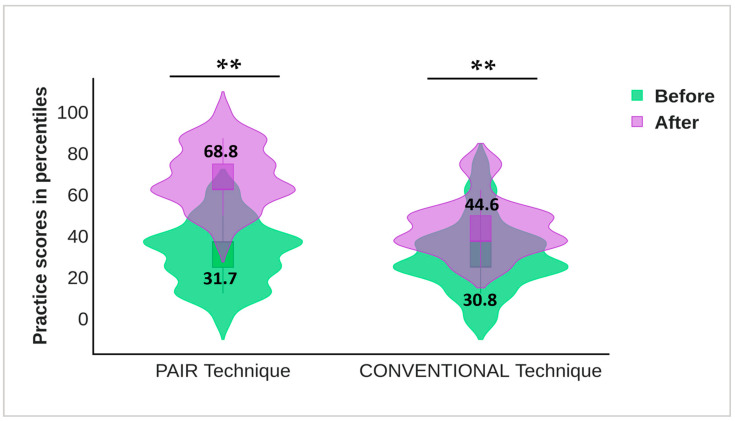
Comparison of practice scores in PAIR group and Conventional group; practice score indicators: unhealthy (≤50th percentile) and healthy (>50th percentile). Baseline and post intervention practice score in both groups has been represented by violin plot. Statistical test used: Paired *t*-test. Level of significance: ** *p* ≤ 0.001 is considered highly statistically significant.

**Table 1 children-10-00369-t001:** Cognitive assessment of the participant according to sociodemographic characteristics.

Cognition	Groups			Gender		
	PAIR *n* (%)	Conventional *n* (%)	*p*-Value	Male *n* (%)	Female *n* (%)	*p*-Value
ASQ (Mean ± SD)	128.47 ± 16.52	127.58 ± 18.14	0.084 ^α^	126.22 ± 17.37	129.37 ± 20.28	0.062 ^α^
VSMS (Mean ± SD)	58.26 ± 12.23	62.37 ± 10.55	0.072 ^α^	55.67 ± 10.79	60.00 ± 3.46	<0.001 ^α,^*
IQ			0.059 ^β^			<0.001 ^β,^**
Intellectually superior	4 (13.33%)	5 (16.67%)		1 (2.63%)	2 (9.09%)	
Definitely above average	4 (13.33%)	3 (10%)		2 (5.26%)	5 (22.7%)	
Intellectually average	12 (40%)	16 (53.33%)		20 (52.63%)	7 (31.82%)	
Definitely below average	9 (30%)	4 (13.33%)		14 (36.84%)	4 (18.18%)	
Intellectually impaired	1 (3.33%)	2 (6.67%)		1 (2.63%)	4 (18.18%)	
Frankl Behavioural scale			0.066 ^β^			0.034 ^β^*
Definitely positive	7 (23.33%)	5 (16.67%)		7 (18.42%)	2 (9.09%)	
Positive	18 (60%)	17 (56.67%)		26 (68.4%)	14 (63.64%)	
Negative	5 (16.67%)	8 (26.67%)		5 (13.16%)	6 (27.27%)	

ASQ: Autism Spectrum Quotient; VSMS: Vineland Social Maturity Scale; IQ: Intelligent Quotient. All values are expressed as mean ± standard deviation (SD) and frequency with percentages (in parentheses). Statistical test used: ^α^ unpaired *t*-test, ^β^ chi-squared test. Level of significance: * *p* ≤ 0.05 for a significant and ** *p* ≤ 0.001 for a highly significant association.

**Table 2 children-10-00369-t002:** Frequency of oral health conditions and their distribution.

Oral Health Conditions	PAIR Group *n* (%) = 30	Conventional Group *n* (%) = 30	Total *n* (%) = 60	*p*-Value
Caries experience—DMFT (Mean ± SD)	2.40 ± 0.12	2.70 ± 0.23	5.10 ± 0.350	0.247 ^ȶ^
Oral hygiene status				0.078 ^ɤ^
Good	5 (16.67%)	8 (26.67%)	13 (21.66%)	
Fair	9 (30%)	7 (23.33%)	16 (26.67%)	
Poor	16 (53.33%)	15 (50%)	31 (51.67%)	
Gingivitis				0.067 ^ɤ^
None or Mild	14 (46.67%)	12 (40%)	26 (43.33%)	
Moderate	9 (30%)	10 (33.33%)	19 (31.67%)	
Severe	7 (23.33%)	8 (26.67%)	15 (25%)	
Dental trauma				0.043 ^ɤ,^*
No trauma	18 (60%)	16 (53.33%)	34 (56.66%)	
Enamel fracture	8 (26.67%)	11 (36.67%)	19 (31.67%)	
Treated injury	4 (13.33%)	3 (10%)	7 (11.67%)	
Oral mucosal lesions				0.025 ^ɤ,^
No abnormal condition	25 (83.34%)	21 (70%)	46 (76.67%)	
Ulceration	4 (13.33%)	6 (20%)	10 (16.66%)	
Soft tissue lesion	1 (3.33%)	3 (10%)	4 (6.67%)	

All values are expressed as mean ± standard deviation (SD) and frequency with percentages (in parentheses). DMFT: Decayed Missing Filled Teeth. Statistical test used: ^ȶ^ unpaired *t*-test, ^ɤ^ chi-squared test. Level of significance: * *p* ≤ 0.05 for a statistically significant.

**Table 3 children-10-00369-t003:** Comparison of caregiver perception of oral hygiene practices in the study population under PAIR technique and conventional technique before and after the intervention.

		Response Frequencies *n* (%)						
Sl. No	Questions		PAIR Technique (*n* = 30)			Conventional Technique (*n* = 30)		
			Pre-Health Education	Post-Health Education	*p*-Value	Pre-Health Education	Post-Health Education	*p*-Value
1.	How does the child clean his/her teeth?	Brush	18 (60%)	26 (86.7%)	<0.001 ^¥,^**	16 (53.3%)	21 (70%)	<0.001 ^¥,^**
		Finger	12 (40%)	4 (13.3%)		14 (46.7%)	9 (30%)	
2.	In which direction does the child brush his/her teeth?	Vertical	7 (23.3%)	0	0.004 ^¶,^**	5 (16.7%)	3 (10%)	0.057 ^¶^
		Horizontal	13 (43.4%)	7 (23.3%)		14 (46.7%)	12 (40%)	
		Circular	4 (13.3%)	23 (76.7%)		3 (10%)	8 (26.7%)	
		Any other	6 (20%)	0		8 (26.6%)	7 (23.3%)	
3.	How many times does the child brush his/her teeth in a day?	Once	21 (70%)	6 (20%)	0.001 ^¶,^**	21 (70%)	17 (56.7%)	<0.001 ^¶,^**
		Twice	5 (16.7%)	16 (53.3%)		4 (13.3%)	9 (30%)	
		After every meal	1 (3.3%)	8 (26.7%)		1 (3.3%)	1 (3.3%)	
		Don’t clean everyday	3 (10%)	0		4 (13.3%)	3 (10%)	
4.	When does the child clean his/her teeth?	Before meal	11 (36.7%)	9 (30%)	0.082 ^¥^	16 (53.3%)	4 (13.3%)	0.063 ^¥^
		After meal	19 (63.3%)	21 (70%)		14 (46.7%)	26 (86.7%)	
5.	What does the child use to clean his/her teeth?	Toothpaste	24 (80%)	29 (96.7%)	0.746 ^¥^	26 (86.7%)	28 (93.3%)	0.951 ^¥^
		Toothpowder	6 (20%)	1 (3.3%)		4 (13.3%)	2 (6.7%)	
6.	How often does the child change their toothbrush?	1–3 months	2 (6.7%)	22 (73.3%)	0.003 ^¶,^**	2 (6.7%)	7 (23.4%)	<0.001 ^¶,^**
		4–6 months	9 (30%)	5 (16.7%)		10 (33.3%)	10 (33.3%)	
		6 months	6 (20%)	3 (10%)		4 (13.3%)	4 (13.3%)	
		Not applicable	13 (43.3%)	0		14 (46.7%)	9 (30%)	
7.	How often does the child rinse his/her mouth with water after eating?	Always	4 (13.3%)	11 (36.7%)	0.369 ^¶^	6 (20%)	9 (30%)	0.448 ^¶^
		Sometimes	9 (30%)	17 (56.6%)		17 (56.7%)	18 (60%)	
		Never	17 (56.7%)	2 (6.7%)		7 (23.3%)	3 (10%)	
8.	Does the child use any other oral hygiene aids?	Yes	2 (6.7%)	9 (30%)	0.527 ^¥^	2 (6.7%)	8 (26.7%)	0.762 ^¥^
		No	28 (93.3%)	21 (70%)		28 (93.3%)	22 (73.3%)	

PAIR technique: Picture Assisted Illustration Reinforcement technique. All values are expressed as the frequency with percentages (in parentheses). Statistical test used: ^¥^ McNemar test and ^¶^ chi-squared test. Level of significance: ** *p* ≤ 0.001 for a highly significant association.

**Table 4 children-10-00369-t004:** Comparison of oral hygiene practice score in the study population of PAIR technique and conventional technique before and after the intervention.

Practice	PAIR Technique (*n* = 30)		Conventional Verbal Technique (*n* = 30)		Statistics	
	Mean ± SD	95% CI	Mean ± SD	95% CI	*t*-Value	*p*-Value ^||^
Before intervention	2.53 ± 1.25	2.07–3.00	2.47 ± 1.38	1.95–2.98	−0.023	0.846
After intervention	5.50 ± 1.12	5.05–5.95	3.57 ± 1.10	3.15–3.98	5.842	<0.001 **
*t*-value	−13.049		−4.387			
95% CI	−3.43 to −2.50		−1.61 to −0.59			
*p*-Value ^§^	<0.001 **		<0.001 **			

PAIR technique: Picture Assisted Illustration Reinforcement technique; CI: confidence interval. All values are expressed as mean ± standard deviation (SD). Statistical test used: ^§^ paired *t*-test, ^||^ unpaired *t*-test. Level of significance: ** *p* ≤ 0.001 is considered highly statistically significant.

## Data Availability

Data available on request to maintain confidentiality. The data presented in this study are available on request from the PI (first author). The data are not publicly available due to detailed information about the children present in the data.

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
