# Peer review of "Evaluation of a “Picture Assisted Illustration Reinforcement” (PAIR) System for Oral Hygiene in Children with Autism: A Double-Blind Randomized Controlled Trial"

_children, 2023, doi:10.3390/children10020369_

Round 1

Reviewer 1 Report

There is no comparison between the characteristics of the two groups at baseline. It is unclear why a comparison was made between all patients divided into two age groups at baseline instead of making a comparison between group A and group B patients.  The statistical analysis used to compare oral hygiene practices (table 3) is not appropriate. In table 2, the results were not reported correctly: in some fields the total of patient is less then 60: why? 

In addition, there are inaccuracies in the text: on line 290 data are reported from table 2, but the reference given is figure 3. the two groups are sometimes referred to as group A and group B, sometimes as group PAIR and group e Coventional: they should be uniform so as not to be confusing.

In discussion, the results of the study are not clearly analysed: for example, the paragraph from line 356 would be better as an introduction rather than a discussion.

Reviewer 2 Report

1.      At the end of your abstract, please provide a "take-home" message.

2.      Put the keywords in a new order based on alphabetical order.

3.      It is unclear whether the author's something new in this work. According to evaluation, several published studies by other researchers in the past adequately explain the issues you made in the present paper. Please be careful to highlight in the introduction section anything really innovative in this work.

4.      To underline the study gaps that the newest research tries to fill, it is crucial to explain the merits, novelty, and limits of earlier studies in the introduction.

5.      Explain specifically the objective of the present study in the last paragraph of the introduction section.

6.      The authors need to explain the previous treatment for children with autism spectrum disorder, there is reducing anxiety level using hug machine as performed by Afif et al. The authors should address this crucial part in the introduction and/or discussion section. Also, to support this explanation, the MDPI-recommended literature should be included as follows: Physiological Effect of Deep Pressure in Reducing Anxiety of Children with ASD during Traveling: A Public Transportation Setting. Bioengineering 2022, 9, 157. https://doi.org/10.3390/bioengineering9040157

7.      To let the reader comprehend the workflow of the current study, the authors could include extra illustrations as a type of figure in the materials and methods rather than simply the main text as a present form.

8.      What is the baseline of patient selection? Is there any protocol, standard, or basis that has been followed? It is unclear since the patient is very heterogeneous with a small number. The resonance involved impacts the present result makes this study flaws. One major reason for rejecting this paper.

9.      It is necessary to provide more information on the manufacturer, country, and specifications of the tools.

10.   The inaccuracy and tolerance of the experimental equipment used in this inquiry are critical details that must be included in the article.

11.   Outcomes must be compared to similar past research.

12.   The discussion in present article is extremely poor in quality as overall. The authors must elaborate on their arguments and provide a thorough justification. Don't just state the results and give a quick explanation.

13.   Please include the limitation of the present study, it is missing.

14.   Add more detail to the conclusion by structuring it as a paragraph rather than in point-by-point as a present form.

15.   In the conclusion, please explain the further research.

16.   The reference should be enriched with literature from the last five years. MDPI reference is strongly recommended.

17.   The authors were encouraged to proofread their work due to grammatical problems and linguistic style.

18.   It is suggested to the authors for providing graphical abstract in the system after revision.

Reviewer 3 Report

Thank you for the opportunity to review this article. However, there are a number of things that require clarification or improvement to make this article worthy of publication:

1. Background is too long. I don't get the urgency to do this research. Try to tell the author what is the background for wanting to do this research. Research gaps are not mentioned in the background. Supporting data is only data regarding caries while the conditions of deficiencies or gaps that occur in the educational process of autistic children are not mentioned.

2. On the result exposure is too long. simply describe the results that are the highlights of the table. The table must be readable by the reader.

3. The discussion does not discuss the results. More explaining theory.

Round 2

Reviewer 2 Report

It has been improved, good job.
